# An Analysis of the Effectiveness and Safety of Upadacitinib in the Treatment of Inflammatory Bowel Disease: A Multicenter Real-World Study

**DOI:** 10.3390/biomedicines13010190

**Published:** 2025-01-14

**Authors:** Hongzhen Wu, Tingting Xie, Qiao Yu, Tao Su, Min Zhang, Luying Wu, Xiaoling Wang, Xiang Peng, Min Zhi, Jiayin Yao

**Affiliations:** 1Department of Gastroenterology, The Sixth Affiliated Hospital, Sun Yat-Sen University, Guangzhou 515000, China; wuhzh27@mail2.sysu.edu.cn (H.W.); sutao3@mail2.sysu.edu.cn (T.S.); zhangm72@mail.sysu.edu.cn (M.Z.); wuly83@mail2.sysu.edu.cn (L.W.); pengx5@mail.sysu.edu.cn (X.P.); 2Guangdong Provincial Key Laboratory of Colorectal and Pelvic Floor Disease, The Sixth Affiliated Hospital, Sun Yat-Sen University, Guangzhou 515000, China; 3Biomedical Innovation Center, The Sixth Affiliated Hospital, Sun Yat-Sen University, Guangzhou 515000, China; 4Digestive Department, The Second Affiliated Hospital of Guangzhou Medical University, Guangzhou 515000, China; jianning.22@163.com; 5The Second Affiliated Hospital, Zhejiang University School of Medicine, Hangzhou 310009, China; yuqiao@zju.edu.cn; 6Department of Clinical Nutrition, The Eighth Affiliated Hospital, Sun Yat-Sen University, Shenzhen 518033, China; wangxling@mail.sysu.edu.cn

**Keywords:** upadacitinib, inflammatory bowel disease, ulcerative colitis, Crohn’s disease

## Abstract

**Background and Aims:** Inflammatory bowel disease (IBD) requires effective treatment options. Upadacitinib, a Janus kinase 1 (JAK1) inhibitor, has shown effectiveness in trials for Crohn’s disease (CD) and ulcerative colitis (UC). This study evaluates its real-world effectiveness and safety. **Methods:** We conducted a multicenter retrospective cohort study in tertiary care centers, involving patients treated with upadacitinib from January 2023 to September 2024. The study included adult patients aged 18 years or older, diagnosed with UC or CD, who received at least 8 weeks of upadacitinib therapy. Treatment outcomes were evaluated using established clinical, endoscopic, imaging, histological, and laboratory parameters. **Results:** A total of 236 IBD patients received upadacitinib treatment. In 80 UC patients at 8 weeks, 64.0% achieved steroid-free remission, 57.6% clinical remission, and 81.8% response. Endoscopic remission was 35.8% (*p* = 0.039), with 63.3% response and 35.8% mucosal healing. Histological remission reached 29.2% (*p* = 0.009). For 156 CD patients at 12 weeks, 76.8% achieved steroid-free remission (*p* < 0.001), 77.8% clinical remission (*p <* 0.001), and 81.0% response. Mean CDAI decreased from 214.9 to 117.5 (*p* < 0.001). Endoscopic remission was 19.4%, with 48.9% response and 4.9% mucosal healing. Radiological remission was 9.1% with 85.7% response. Intestinal ultrasound showed 5.7% remission and 56.7% response. **Conclusions:** Upadacitinib demonstrates significant real-world effectiveness and safety in IBD, particularly in biologic-resistant cases, as evidenced by high rates of steroid-free remission and clinical response. These outcomes are likely due to its targeted JAK1 inhibition, which effectively reduces inflammation and promotes mucosal healing. Future research should focus on long-term safety, comparative effectiveness with other biologics, and its application in diverse patient populations. These findings support the integration of upadacitinib into IBD management strategies.

## 1. Introduction

Inflammatory bowel disease (IBD), encompassing Crohn’s disease (CD) and ulcerative colitis (UC), represents a group of idiopathic disorders characterized by chronic inflammation of the gastrointestinal tract. The pathogenesis of IBD is complex, involving genetic, environmental, and immunological factors, leading to a dysregulated immune response [1]. Despite advancements in understanding the underlying mechanisms, the management of IBD remains a significant clinical challenge. Traditional therapeutic strategies, including corticosteroids, immunomodulators, and biologic agents, have afforded many patients relief from the debilitating symptoms of IBD.

However, a substantial proportion of patients either fail to respond adequately or lose response over time to these conventional treatments, underscoring the need for novel therapeutic options [2]. In recent years, there has been a trend in drug development favoring small molecule drugs, which are advantageous due to their oral administration, potential for cost savings, and quick action. Janus kinase (JAK) inhibitors have emerged as a promising class of small molecule therapeutics due to their roles in modulating cytokine signaling pathways, which are pivotal in IBD pathogenesis.

The JAK/STAT signaling pathway is crucial in the pathophysiology of inflammatory bowel disease (IBD), mediating the effects of over 50 cytokines, including interleukins, interferons, and growth factors. This pathway involves four JAK family members (JAK1, JAK2, JAK3, and TYK2) that transduce signals through various STAT proteins, regulating hematopoiesis and immune cell functions and contributing to the inflammatory environment observed in IBD. Upadacitinib, a selective JAK1 inhibitor (C_17_H_19_F_3_N_6_O, molecular weight ~380.4 g/mol), achieves high specificity by inhibiting JAK1 with 40-, 130-, and 190-fold greater potency than JAK2, JAK3, and TYK2, respectively. Pharmacodynamically, upadacitinib competitively binds to the ATP-binding site of JAK1, inhibiting STAT phosphorylation and subsequent gene regulation involved in inflammation, effectively suppressing pro-inflammatory cytokine production while minimizing off-target effects [3,4,5]. The detailed signaling pathway process is illustrated in Figure 1.

Structural modifications enable reversible binding to JAK1, allowing for rapid dissociation and reducing the risk of long-term adverse reactions. Pharmacokinetically, upadacitinib exhibits linear characteristics with minimal plasma accumulation, is primarily excreted unchanged via hepatic metabolism with CYP3A4 playing a minor role, and has a half-life of approximately 8–14 h, supporting once-daily dosing. The detailed molecular structure and 3D conformation of upadacitinib are shown in Figure 2 [6]. These attributes make upadacitinib a promising therapeutic agent for IBD, addressing the limitations of existing treatment options.

Initially approved for rheumatoid arthritis, upadacitinib has demonstrated effectiveness and an acceptable safety profile in this context. Clinical studies in CD and UC have shown its potential to reduce inflammatory markers and improve clinical outcomes. However, there is a pressing need to evaluate the results from significant multicentric retrospective studies to firmly establish its role. Comparing clinical trial outcomes with real-world data is imperative to understand upadacitinib’s application in routine clinical practice. The long-term safety data, particularly, remains a significant concern and a gap in the literature.

Compared to previous studies, our multicentric retrospective cohort study offers several strengths that address the limitations observed in earlier research. While prior investigations often focused on single-center data with smaller sample sizes and shorter follow-up periods, our study encompasses multiple tertiary care centers, enhancing the generalizability of the findings. Additionally, we included a larger and more diverse patient population, allowing for a comprehensive analysis of upadacitinib’s effectiveness and safety across different demographics and disease phenotypes. Unlike some previous studies that primarily relied on clinical outcomes, our research integrates a multifaceted assessment approach, incorporating endoscopic, imaging, histological, and laboratory parameters to provide a holistic evaluation of treatment response. These robust methodologies and broader scope position our study as a more definitive exploration of upadacitinib’s role in IBD management, underscoring its potential for publication and contribution to the field.

## 2. Methods

### 2.1. Data Selection

This multicenter retrospective cohort study collected data from IBD patients treated with upadacitinib at three Chinese hospitals between 1 January 2023, and September 2024. Patients were diagnosed using established criteria, including clinical symptoms, biomarkers, endoscopic, histological, and radiological findings. The study was approved by each hospital’s Medical Ethics Committee (Ethics Approval Numbers: 2024ZSLYEC-063, Res2024-0040, 2024-hg-ks-08) and registered at ClinicalTrials.gov (NCT06274996). Patient confidentiality was maintained, and informed consent was waived due to the retrospective nature. Data were obtained from electronic medical records by staff at each center.

### 2.2. Inclusion and Exclusion Criteria

Inclusion criteria: (1) hospitalized patients at our center and two other centers from January 2023 to September 2024; (2) diagnosed with CD or UC; (3) aged 18 years or older; (4) received at least 8 weeks of upadacitinib treatment by September 2024. Exclusion criteria: (1) pregnancy and breastfeeding; (2) serious infections or history of malignancy; (3) active or chronic infections; (4) history of cardiovascular diseases, including coronary artery disease, heart failure, and stroke; (5) presence of cardiovascular risk factors, such as uncontrolled hypertension (systolic blood pressure ≥ 140 mmHg or diastolic blood pressure ≥ 90 mmHg), diabetes mellitus (HbA1c ≥ 7.0%), obesity (BMI ≥ 30 kg/m^2^), or metabolic syndrome; (6) liver impairment (ALT or AST > 3 times the upper limit of normal) or kidney impairment (estimated glomerular filtration rate < 30 mL/min/1.73m^2^); (7) history of total colectomy for UC; (8) history of venous or arterial thrombosis, or presence of thrombotic risk factors such as inherited or acquired thrombophilia.

### 2.3. Investigated Drugs

For both patients with UC or CD, upadacitinib was administered according to the manufacturer’s instructions. Patients with CD received an induction dose of 45 mg once daily for 12 weeks, while those with UC were given the same dose once daily for 8 weeks. This dosing regimen aligns with the established protocols for upadacitinib induction therapy in IBD [7,8].

### 2.4. Demographic and Clinical Data

Baseline and follow-up data were extracted from electronic medical records by two physicians at each center. One collected data using a standardized form, while the other verified it. Discrepancies were resolved through discussion to reach consensus.

Collected data included demographic information (gender, age, age at diagnosis, BMI), disease characteristics (duration, location, behavior, perianal disease, extraintestinal manifestations), smoking status, concomitant and previous therapies, and clinical outcomes. For UC, we assessed clinical remission, endoscopic outcomes (Mayo endoscopic subscore, UCEIS), and histologic remission. For CD, we evaluated clinical remission, endoscopic outcomes (SES-CD), and imaging outcomes, including radiological remission and Intestinal Doppler Ultrasound parameters. Laboratory parameters such as albumin, hemoglobin, CRP, ESR, platelet count, liver function tests, and D-dimer were also recorded. Fecal calprotectin was not routinely collected due to unavailability and out-of-pocket costs.

In accordance with the 2023 Chinese IBD guidelines and STRIDE II, early mucosal healing leads to better long-term outcomes [9]. Thus, we recommend endoscopic evaluation post induction to assess mucosal healing, which is standard practice at our centers with well-established follow-up systems. While some patients missed endoscopies due to compliance issues, comparing baseline characteristics (age, gender, disease duration, location, activity) and outcomes (clinical remission, adverse events) between those with and without endoscopies revealed no significant differences, suggesting minimal selection bias (Appendix A).

Disease activity in UC patients was evaluated using the modified Mayo score, with clinical remission defined as a score ≤ 2 and no single item score > 1. Clinical response was defined as a ≥ 30% decrease from baseline, a score ≥ 3, and a ≥ 1 decrease in bleeding score or a score of 0–1. The endoscopic assessment included both Mayo endoscopic subscore and UC Endoscopic Index of Severity (UCEIS). Endoscopic remission was defined as a Mayo endoscopic subscore of 0–1, and mucosal healing was defined as a Mayo endoscopic subscore of 0, while endoscopic response was defined as a ≥ 1 decrease from baseline. Histological remission was evaluated using the Geboes scoring system. For CD assessment, CDAI was used to evaluate disease severity, with clinical remission defined as CDAI < 150 and clinical response as a ≥ 70 decrease. SES-CD was used for endoscopic evaluation, with scores ≤ 2 indicating remission and a ≥50% decrease indicating response. Mucosal healing was defined as absence of ulceration with only mild erythema (SES-CD 0–2). Perianal disease severity was assessed using VanAssche scoring based on anal MRI. Radiological response was evaluated through CT/MR enterography, defined by improvements in BWT, inflammatory fat, mural blood flow, and enhancement, with remission defined as the normalization of these parameters. The intestinal ultrasound assessment showed response when BWT decreased, and remission when wall thickness normalized (≤3 mm for small bowel, ≤4 mm for colon) without complications (strictures, penetration, fistulas, adhesions, or abscesses). IUS-SAS scoring provided a comprehensive ultrasound evaluation. Laboratory parameters, including inflammatory markers, nutritional status, and liver and kidney function, were systematically monitored in both UC and CD patients. Adverse events were documented to evaluate treatment safety.

### 2.5. Outcome

The primary outcomes were steroid-free clinical remission rate, clinical remission rate, and clinical response rate at week 8 for UC and week 12 for CD patients. Secondary endpoints included endoscopic remission and response rates, mucosal healing rate, histological remission rate (Geboes score for UC), and radiological outcomes. For CD, the radiological assessment comprised CT/MR enterography and intestinal ultrasound findings, with ultrasound evaluation including BWT normalization and IUS-SAS scoring. The adverse events monitored included leukopenia, liver function abnormalities, infections (tuberculosis and herpes zoster), malignancies, intestinal perforation, non-melanoma skin cancer, cardiovascular events, dyslipidemia, hospitalizations, and death. Serious adverse events were defined as intestinal perforation, cardiovascular events, hospitalizations, or death.

### 2.6. Statistical Analysis

Data analysis was conducted using SPSS 22.0 (IBM Corp., Armonk, NY, USA). For quantitative variables (ALB, HGB, CRP, ESR), normality was assessed by the visual inspection of histograms and Q-Q plots, considering the limitations of formal normality tests in small sample sizes. Data were presented as mean ± SD or median and IQR, based on the observed distribution. The choice of statistical tests was guided by the distribution characteristics of the data. Repeated measures within subjects over time were analyzed using either paired *t*-tests for normally distributed data or the Wilcoxon signed-rank test for non-normal data. Paired *t*-tests were selected for their ability to compare means in normally distributed paired samples, while the Wilcoxon signed-rank test was used for its non-parametric nature, suitable for data that do not follow a normal distribution. Between-group differences were determined using independent *t*-tests or the Mann–Whitney U test, based on the data’s distribution and homogeneity of variances, confirmed by Levene’s test. Independent *t*-tests were employed to compare means between two independent groups when data were normally distributed and variances were equal, whereas the Mann–Whitney U test was used for comparing medians between groups when these assumptions were not met. Patients with >30% missing data were excluded from the analysis. For missing endoscopic data, sensitivity analyses were performed to exclude bias. In cases where a large proportion of data were missing (>50%), the data were objectively presented without further processing. The number of patients with missing data and the number of patients who underwent endoscopic evaluation are reported in the Results section.

All authors had access to the study data and reviewed and approved the final manuscript.

## 3. Results

### 3.1. Patient Selection

We evaluated the effectiveness of upadacitinib in 236 patients with UC or CD from both inpatient and outpatient departments. During the study period, 17 patients were excluded due to missing baseline and 8/12-week data. Additionally, 18 patients were excluded due to surgical needs, treatment modifications, inadequate response, or financial constraints. The final analysis included 201 patients (71 UC and 130 CD). All patients initiated treatment with an FDA-approved induction dose of 45 mg/day. The overall process is shown in Figure 3. The specific data were shown in Table 1.

### 3.2. Ulcerative Colitis, UC

An 80-patient cohort study with UC was conducted, where patients were treated with upadacitinib for a minimum of 8 weeks. The mean age was 43.7 years (SD: 14.6), with a median disease duration of 2.0 years (IQR: 0.3–6.0 years). The cohort was 47.5% male, with a mean BMI of 24.93 kg/m^2^ (SD: 20.4). Disease location was stratified according to Montreal classification: 67.3% with E3 (extensive colitis), 22.4% with E2 (left-sided colitis), and 10.2% with E1 (proctitis). A total of 11.4% of patients had perianal disease, and various extraintestinal manifestations were observed: articular manifestations (7.6%), oral ulcers (5.1%), and skin rash (2.5%). The majority (96.2%) had never smoked, with 3.8% being ex-smokers. Prior treatment history included biologics (83.5%), glucocorticoids (69.6%), immunosuppressive agents (27.8%), and tofacitinib (11.4%). Current concomitant therapies comprised EEN (29.0%), glucocorticoids (19.4%), VDZ (16.1%), 5-ASA (14.0%), and IFX (8.1%). Prior to upadacitinib treatment, 95.3% of the cohort had received biologic therapy, 27.9% immunosuppressants, and 79.1% corticosteroids, with all patients having used 5-ASA. A minority, 4.7%, had experience with EEN, and 16.3% had been treated with tofacitinib. Concomitant therapies during the study included corticosteroids (23.3%), 5-ASA (14.0%). Three patients (7.0%) received dual advanced therapy with upadacitinib and vedolizumab at the initiation of upadacitinib treatment. These patients had a history of inadequate response to TNF inhibitors and were considered to have treatment-refractory UC, necessitating a dual-targeted approach. Treatment optimization was reported for 23.3% of patients before the study.

At the 8-week follow-up, steroid-free clinical remission increased from 2.8% (1/36) to 64.0% (17/27). Clinical remission improved from 4.4% (2/45) to 57.6% (19/33), with 81.8% (27/33) achieving clinical response. The mean modified Mayo score significantly decreased from 6.9 ± 2.1 to 3.1 ± 2.5 (*p* = 0.001). Endoscopic outcomes showed improvement, with endoscopic remission increasing from 4.4% (2/45) to 35.8% (19/53) (*p* = 0.039), and 63.3% (31/49) achieving endoscopic response. Mucosal healing improved from 2.3% (1/44) to 35.8% (9/25). Both Mayo endoscopic subscore (2.7 ± 0.6 to 2.1 ± 1.1, *p* = 0.001) and UCEIS (5.3 ± 1.6 to 4.1 ± 2.7, *p* = 0.009) showed significant improvements. Notably, histologic remission increased from 0% to 29.2% (7/24) (*p* = 0.009).

Nutritional parameters improved significantly, with mean ALB levels increasing from 37.0 ± 5.8 g/L to 40.3 ± 5.8 g/L (*p* = 0.001), and the proportion of patients with normal ALB rising from 63.0% (41/65) to 86.3% (44/51) (*p* = 0.001). Mean HGB levels increased from 110.2 ± 24.0 g/L to 114.7 ± 35.5 g/L, with normal HGB rates improving from 43.1% (31/72) to 56.1% (32/57). Inflammatory markers showed improvement, with normal CRP rates increasing from 51.4% (36/70) to 77.8% (42/54) (*p* = 0.001), while normal ESR rates remained stable from 62.1% (36/58) to 65.1% (28/43). Other laboratory parameters, including PLT, liver function tests, and renal function markers, were maintained within normal ranges throughout the treatment period. One case of asymptomatic pulmonary embolism was reported in a 60-year-old female patient treated with upadacitinib. The event was incidentally detected during routine examinations when the patient experienced disease relapse in the active phase of UC. The patient did not present with any clinical symptoms suggestive of pulmonary embolism. Following the diagnosis, appropriate anticoagulant therapy was initiated. A follow-up chest CT showed a significant reduction in the pulmonary embolism compared to the initial findings (detailed information available in the Appendix A). The patient’s condition remained stable without further complications. Other common adverse events were reported, including menorrhagia, acne, lateral leg paresthesia, arthralgia, and dermatological manifestations (alopecia and papular rash). No other adverse events, including serious infections, malignancies, gastrointestinal perforations, or deaths, were observed in the UC group throughout the study period. The results of this analysis are plotted in Figure 4 and Figure 5 and Appendix A. The specific data are shown in Table 2.

### 3.3. Crohn’s Disease, CD

A total of 156 CD patients were treated with upadacitinib, with a mean age of 30.1 ± 9.98 years and a median disease duration of 5.0 years (IQR: 3.0–8.0). Males comprised 73.1% of the cohort, and the mean BMI was 19.9 ± 3.1 kg/m^2^. According to the Montreal classification, disease onset age was A1 in 6.2%, A2 in 80.1%, and A3 in 13.7%. Disease localization was ileal (L1) in 15.1%, colonic (L2) in 9.4%, ileocolonic (L3) in 74.5%, and L3 + L4 in 0.9%. Disease behavior showed inflammatory phenotype (B1) in 38.1%, stricturing (B2) in 43.8%, penetrating (B3) in 16.2%, and combined B2 + B3 in 1.9%. Perianal disease was present in 72.3% of patients. Extraintestinal manifestations were observed in 21.6% of patients, including oral ulcers (12.8%), articular manifestations (5.8%), and skin rash (1.9%). Regarding smoking status, 94.6% were never smokers, 3.4% were ex-smokers, and 2.0% were current smokers. All patients had previous exposure to biologics, with 60.1% receiving immunosuppressive agents, 41.7% glucocorticoids, 63.5% EEN, and 5.4% tofacitinib. Current concomitant treatments included glucocorticoids (4.0%) and EEN (24.8%). Combined biological agents included IFX (1.9%), ADA (3.2%), UST (11.0%), and VDZ (0.6%).

In the 12-week follow-up of 156 CD patients treated with upadacitinib, the proportion of patients achieving steroid-free clinical remission increased from 19.4% (20/103) to 76.8% (61/79), and clinical remission improved from 19.8% (21/106) to 77.8% (63/81) (both *p* < 0.001). Clinical response was observed in 81.0% (64/79) of patients at week 12. The mean CDAI score decreased significantly from 214.9 ± 77.9 to 117.5 ± 79.3 (*p* < 0.001). Endoscopic assessments showed that endoscopic remission increased from 12.6% (12/95) to 19.4% (12/62) with 48.9% (23/47) achieving endoscopic response. Mucosal healing rates were comparable between baseline and week 12 (5.3% vs. 4.9%). The median Simple Endoscopic Score for Crohn’s Disease (SES-CD) score decreased from 10 (5.0–18.0) to 4 (1.0–8.5) (*p* < 0.001). Radiological evaluation demonstrated improvements in remission rates from 1.1% (1/91) to 9.1% (2/22) (*p* < 0.001), with 85.7% (18/21) achieving radiological response at week 12. Nutritional markers showed significant enhancement, with mean albumin levels increasing from 37.3 ± 5.5 g/L to 41.4 ± 5.0 g/L (*p* < 0.001). Inflammatory burden improved, as evidenced by an increase in the proportion of patients with normal CRP from 60.0% (90/150) to 82.7% (81/98) (*p* < 0.001). The proportion of patients with normal ESR increased from 62.0% (75/121) to 78.6% (55/70), though not statistically significant.

Adverse events were documented during the treatment period. The most common adverse event was acne (66.7%), followed by gastrointestinal symptoms, including stomach pain (11.1%), headache (5.6%), anemia (5.6%), folliculitis (5.6%), and herpes zoster (5.6%). Most adverse events were mild and did not lead to treatment discontinuation. The results of this analysis are plotted in Figure 6 and Figure 7 and Appendix A. The specific data are shown in Table 3.

## 4. Discussion

In this study, we present evidence that upadacitinib effectively treats both UC and CD in a Chinese cohort, demonstrating significant improvements across multiple disease parameters. For UC patients, the achievement of steroid-free clinical remission (64.0%) at week 8, alongside substantial improvements in Mayo scores, UCEIS, and histological remission (29.2%), validates the drug’s effectiveness at both clinical and tissue levels. In CD patients, the 12-week outcomes showed remarkable improvements, with 76.8% achieving steroid-free clinical remission and significant reductions in SES-CD scores. The comprehensive evaluation through multiple modalities, including endoscopy, histology, and imaging (particularly in perianal disease and intestinal ultrasound scores), provides robust evidence of therapeutic effectiveness. The favorable safety profile and significant improvements in nutritional and inflammatory markers support upadacitinib as a valuable addition to the IBD treatment arsenal, particularly for patients resistant to current therapies, contributing to the advancement of personalized medicine in this domain.

Upadacitinib, a Janus kinase 1 (JAK1) inhibitor, is approved for moderate to severe UC and treatment-resistant CD. Data from three phase 3 clinical trials—two induction studies and one maintenance study—underpin its effectiveness [10]. Upadacitinib was effective at week 8 (induction) and week 52 (maintenance) in patients with moderate to severe UC, notably in those previously treated with TNF antagonists. In cases of moderate to severe CD, the drug achieved endoscopic remission by 16 weeks, with sustained effects up to 52 weeks [11]. Our research indicates higher clinical remission and response rates in IBD patients treated with upadacitinib than with prior biological therapies [12], suggesting a potential for superior clinical remission rates in this refractory patient population.

Upadacitinib offers rapid symptom relief for patients with UC and CD, with marked symptom improvement observed at 2, 4, and 8 weeks following the initiation of treatment [13]. This encompasses a reduction in stool frequency, rectal bleeding, abdominal pain, and urgency of defecation. These advantages, surpassing those reported for ustekinumab and vedolizumab, contribute to an improved quality of life as early as the first two weeks of treatment [13]. The prompt effects of upadacitinib are further supported by significant decreases in inflammation markers; CRP levels show substantial reductions by the 8-week mark in UC and by 12 weeks in patients with CD. These findings suggest that upadacitinib not only mitigates symptoms but also rapidly exerts anti-inflammatory effects in moderate to severe IBD cases.

In the management of IBD, the fundamental aim is to control symptoms and, in pediatric cases, to restore growth. Long-term goals include achieving endoscopic healing and improving the quality of life. Consequently, monitoring treatment outcomes via endoscopic imaging is essential [10]. Phase 3 trials have demonstrated upadacitinib’s effectiveness in attaining histological and endoscopic mucosal improvements in UC, as measured by the Histologic Endoscopic Mucosal Improvement (HEMI) index, and in mucosal healing [14]. Furthermore, a phase 2 trial in CD indicated that upadacitinib led to higher endoscopic remission rates than placebo, a finding that is corroborated by real-world evidence and meta-analyses [15,16]. In our research, significant improvements in endoscopic and histologic outcomes were observed at week 8, with substantial increases in endoscopic remission rates and marked reductions in both Mayo endoscopic subscore and UCEIS, alongside the achievement of histologic remission in previously unresponsive patients. For patients with CD, evaluation at week 12 revealed meaningful endoscopic improvements, particularly evidenced by endoscopic response in nearly half of the patients and a significant reduction in SES-CD scores, though changes in endoscopic remission and mucosal healing rates were less pronounced compared to the UC group.

Radiological assessments are essential in evaluating treatment outcomes for patients with IBD, offering valuable insights into the extent and severity of the disease. These assessments are critical for monitoring both disease activity and the effectiveness of therapeutic interventions. Furthermore, they are key in identifying complications, including perforation, obstruction, and inter-intestinal fistula [17]. Our study’s findings demonstrated marked improvements in radiological outcomes among CD patients at week 12, with a high proportion achieving radiological response. The Intestinal Doppler Ultrasound evaluation showed significant improvement in disease activity, reflected by reduced IUS-SAS scores. Additionally, patients with perianal involvement showed radiological improvement as evidenced by decreased VanAssche scores.

In the clinical management of CD and UC, malnutrition and nutrient deficiencies are frequently encountered. Nutritional interventions can enhance patients’ nutritional status and potentially benefit inflammatory activity, which in turn may impact treatment response and prognosis [18]. Our follow-up data indicate increases in albumin (ALB) levels at 8 weeks post intervention for patients with UC and at 12 weeks for patients with CD. However, no significant changes were observed in Hb levels in either UC or CD patients during the follow-up period.

Upadacitinib’s treatment course has shown a generally favorable safety profile but comes with risks. Previous studies have reported safety concerns for upadacitinib induction therapy, including herpes simplex virus infections and cardiovascular events, with pulmonary embolism and deep vein thrombosis occurring 26 days post-treatment cessation. Liver disease and neutropenia are also potential adverse effects [14,19]. In our cohort, there was one case of pulmonary artery thrombosis in a UC patient, suggesting a need to consider JAK inhibitors’ effects on hemostasis and platelet function. The most common adverse events across both UC and CD patients included acne, gastrointestinal symptoms, and dermatological manifestations. Other reported adverse events were headache, paresthesia, arthralgia, anemia, and herpes zoster.

Biologic agents like anti-TNFα (infliximab, adalimumab), anti-integrin (vedolizumab), and anti-IL-12/23 (ustekinumab) are the standard for moderate to severe IBD. However, they have limitations, such as non-response, loss of responsiveness, infection risks, immunogenicity, and high costs [15]. Janus kinase (JAK) inhibitors, especially upadacitinib, and tofacitinib, are emerging alternatives that target the JAK-STAT pathway. This pathway is crucial for cellular functions, including immunity and hematopoiesis. Upadacitinib, with its short half-life, rapid action, oral administration, and low immunogenicity, improves convenience and adherence. Its selectivity for JAK1 over other JAK kinases may enhance clinical outcomes [15]; however, further research is required to confirm long-term benefits.

This is the first study from China to investigate the real-world use of upadacitinib in IBD treatment. Our multicenter study contributes significant insights into the use of upadacitinib and its impact on Chinese patients with IBD, given the scarcity of available data. Our centers, dedicated to evaluating the clinical effectiveness of novel IBD treatments, employ a rigorous approach to patient follow-up, ensuring the collection of high-quality data and adherence to treatment protocols. Our findings enhance the understanding of upadacitinib’s clinical outcomes, effectiveness, and safety in a real-world setting. The structured follow-up has been crucial for monitoring long-term outcomes and assessing drug tolerability, providing preliminary yet crucial data for future research. This study aids in clinical decision-making and sets the stage for developing strategies to optimize IBD treatment on a global scale.

Our study is limited by its retrospective design, which does not establish causality as effectively as a randomized controlled trial would. To gather more robust evidence, future research should employ prospective, randomized designs. The brevity of the follow-up period is attributable to the recent introduction of upadacitinib, which underscores the necessity for data on long-term effectiveness and safety. While our sample size is adequate, it may not fully reflect the diversity of the broader IBD patient population. Nevertheless, our findings provide an important contribution to the clinical evidence base for upadacitinib, and larger, future studies are needed to validate and expand upon our results.

In summary, our real-world study endorses upadacitinib’s effectiveness and rapid onset in treating refractory IBD, hinting at its potential as an alternative for biologic-resistant patients. Its acceptable safety profile, with manageable adverse effects, bolsters its viability as a new option for complex IBD cases. Our findings enrich the evidence base, positioning upadacitinib as a viable contender in the therapeutic landscape for difficult-to-manage IBD, meriting consideration in treatment protocols.

## 5. Conclusions

Upadacitinib demonstrates significant real-world effectiveness and safety in treating both Crohn’s disease and ulcerative colitis. The high rates of steroid-free remission and clinical response observed in this multicentric retrospective cohort study underscore upadacitinib’s potential as a valuable treatment option, particularly for patients resistant to conventional biologic therapies. The comprehensive assessment, including clinical, endoscopic, histological, and imaging parameters, provides robust evidence supporting its use in diverse patient populations.

Future research endeavors should focus on long-term safety and efficacy, comparative studies with other emerging biologic and small molecule therapies, and the exploration of biomarkers to predict patient response to upadacitinib. Additionally, investigating the drug’s impact on quality of life and its role in combination therapy regimens could further enhance treatment strategies for IBD. These future studies will help in fully defining upadacitinib’s therapeutic role and optimizing its use in clinical practice.

## Figures and Tables

**Figure 1 biomedicines-13-00190-f001:**
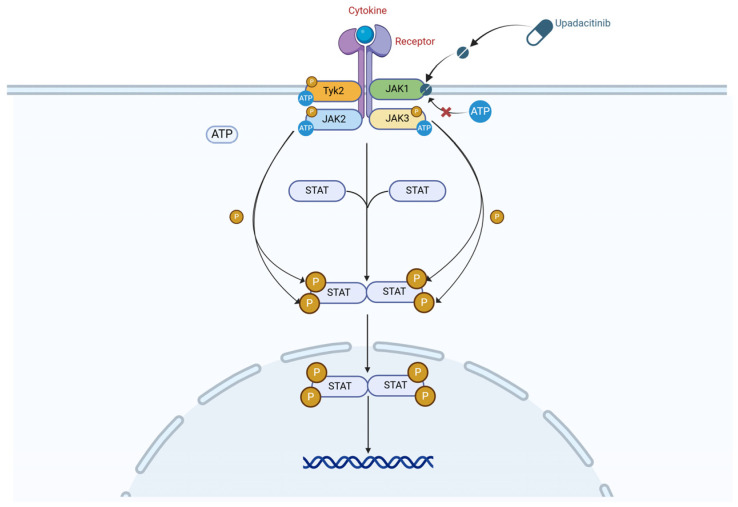
The JAK/STAT signaling pathway and the selective inhibition of JAK1 by upadacitinib in inflammatory bowel disease. JAK1, Janus kinase 1; JAK2, Janus kinase 2; JAK3, Janus kinase 3; TYK2, Tyrosine kinase 2; STAT, Signal Transducer and Activator of Transcription; ATP, Adenosine Triphosphate; P, phosphate group.

**Figure 2 biomedicines-13-00190-f002:**
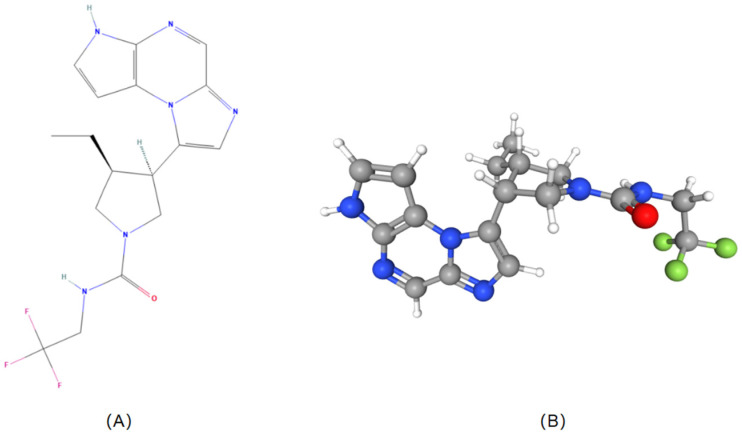
Molecular structure (**A**) and 3D conformation (**B**) of upadacitinib.

**Figure 3 biomedicines-13-00190-f003:**
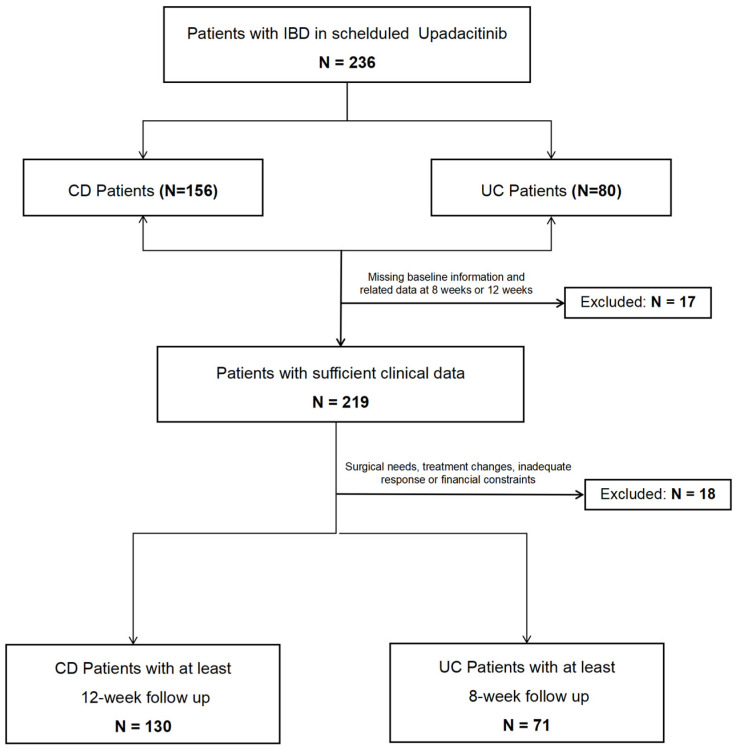
Patient selection flow chart.

**Figure 4 biomedicines-13-00190-f004:**
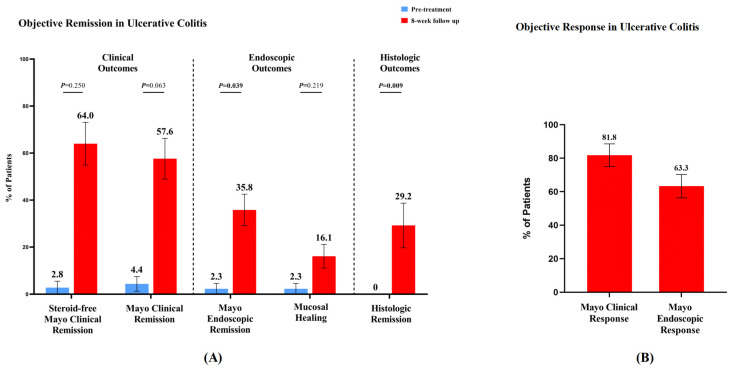
(**A**) Comparison of clinical, endoscopic, and histologic remission rates in ulcerative colitis patients before and after 8 weeks of upadacitinib treatment. (**B**) Clinical response rates in patients with ulcerative colitis after upadacitinib treatment.

**Figure 5 biomedicines-13-00190-f005:**
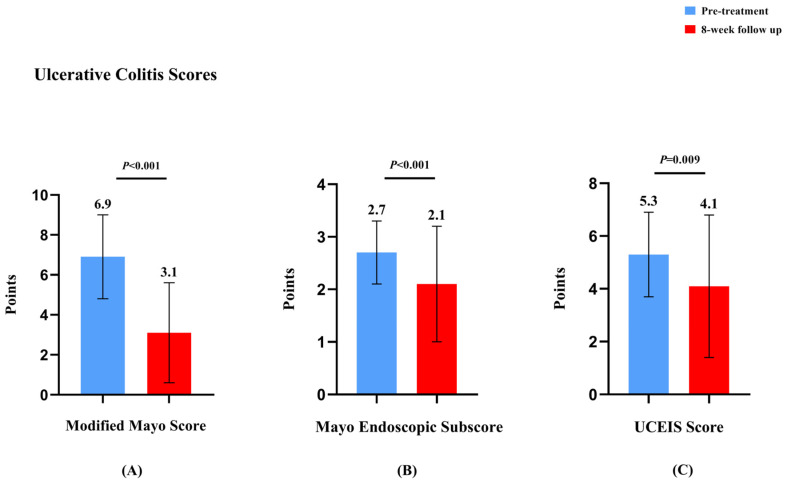
Comparison of modified Mayo, Mayo endoscopic subscore, and UCEIS in ulcerative colitis patients before and after 8 weeks of upadacitinib treatment. (**A**) Comparison of modified Mayo scores in ulcerative colitis patients before and after 8 weeks of upadacitinib treatment. (**B**) Comparison of Mayo endoscopic subscores in ulcerative colitis patients before and after 8 weeks of upadacitinib treatment. (**C**) Comparison of UCEIS scores in ulcerative colitis patients before and after 8 weeks of upadacitinib treatment.

**Figure 6 biomedicines-13-00190-f006:**
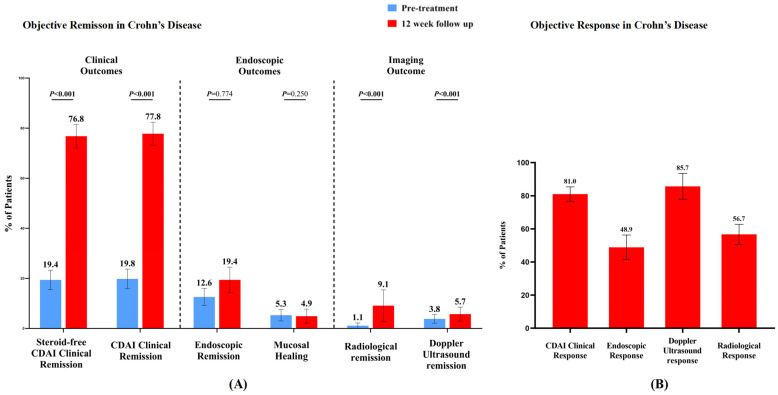
(**A**) Comparison of clinical, endoscopic, and imaging remission rates in Crohn’s patients before and after 12 weeks of upadacitinib treatment. (**B**) Clinical, endoscopic, radiological, and imaging response rates in Crohn’s patients before and after 12 weeks of upadacitinib treatment.

**Figure 7 biomedicines-13-00190-f007:**
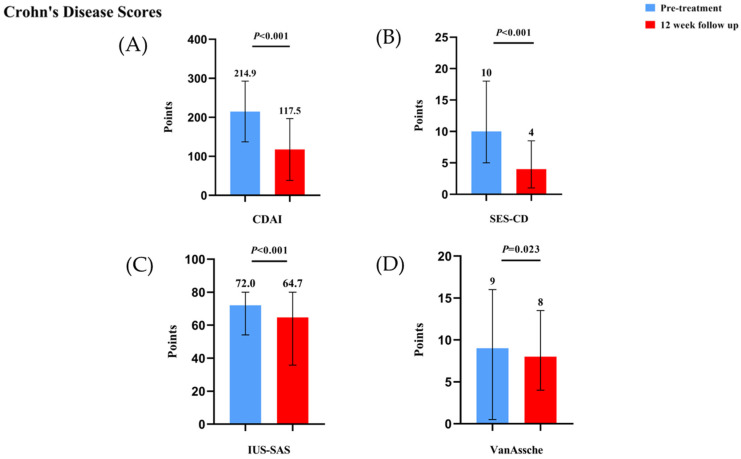
(**A**) Comparison of CDAI scores in Crohn’s disease patients before and after 12 weeks of upadacitinib treatment. (**B**) Comparison of SES-CD scores in Crohn’s disease patients before and after 12 weeks of upadacitinib treatment. (**C**) Comparison of IUS-SAS scores in Crohn’s disease patients before and after 12 weeks of upadacitinib treatment. (**D**) Comparison of VanAssche scores in Crohn’s disease patients before and after 12 weeks of upadacitinib treatment.

**Table 1 biomedicines-13-00190-t001:** Baseline characteristics of patients with IBD.

Variables	CD Patients (N = 156)	UC Patients (N = 80)
Male (n, %)	114 (73.1)	38 (47.5)
BMI (kg/m^2^, mean ± SD)	19.9 ± 3.1	24.93 ± 20.4
Age (years, mean ± SD)	30.1 ± 9.98	43.7 ± 14.6
Disease duration (years, median (IQR))	5.0 (3.0–8.0)	2.0 (0.3–6.0)
A1	9 (6.2)	-
A2	117 (80.1)	-
A3	20 (13.7)	-
Disease location (CD) (n, %)		
L1	16 (15.1)	-
L2	10 (9.4)	-
L3	79 (74.5)	-
L3 + L4	1 (0.9)	-
Disease location (UC) (n, %)		
E1	-	5 (10.2)
E2	-	11 (22.4)
E3	-	33 (67.3)
Disease behavior (CD) (n, %)		
B1	40 (38.1)	-
B2	46 (43.8)	-
B3	17 (16.2)	-
B2 + B3	2 (1.9)	
Perianal disease (n, %)	107 (72.3)	9 (11.4)
Extraintestinal manifestation (n, %)	8 (21.6)	1 (2.3)
Oral ulcers	20 (12.8)	4 (5.1)
Articular manifestations	9 (5.8)	6 (7.6)
Skin rash	3 (1.9)	2 (2.5)
Smoking status (n, %)		
Never smoked	141(94.6)	76 (96.2)
Ex-smoker	5 (3.4)	3 (3.8)
Current smoker	3 (2.0)	0 (0.0)
Concomitant therapy (n, %)		
Immunosuppressive agents	0 (0.0)	0 (0.0)
Glucocorticoids	6 (4.0)	12 (19.4)
5-ASA	0 (0.0)	6 (14.0)
EEN	37 (24.8)	18 (29.0)
Combined use of biological agents (n, %)		
IFX	3 (1.9)	5 (8.1)
ADA	5 (3.2)	0 (0.0)
UST	17 (11.0)	0 (0.0)
VDZ	1 (0.6)	10 (16.1)
History of previous therapy (n, %)		
Immunosuppressive agents	89 (60.1)	22 (27.8)
Glucocorticoids	65 (41.7)	55 (69.6)
EEN	99 (63.5)	11 (13.9)
Biologics	156 (100.0)	66 (83.5)
Tofacitinib	8 (5.4)	9 (11.4)

**BMI**, Body Mass Index; **CD**, Crohn’s disease; **UC**, ulcerative colitis; **5-ASA**, 5-Aminosalicylic Acid; **EEN**, Exclusive Enteral Nutrition; **IFX**, infliximab; **ADA**, adalimumab; **UST**, ustekinumab; **VDZ**, vedolizumab; **IQR**, Interquartile Range.

**Table 2 biomedicines-13-00190-t002:** Pre-treatment and 8-week follow-up clinical, endoscopic, and laboratory outcomes of upadacitinib treatment in patients with UC.

Outcomes	Pre-Treatment (N = 80)	8-Week Follow Up (N = 71)	*p*-Value
Clinical Outcomes			
Steroid-free clinical remission (n, %)	1 (2.8)	17 (64.0)	0.250
Clinical remission (n, %)	2 (4.4)	19 (57.6)	0.063
Clinical response (n, %)		27 (81.8)	
Modified Mayo score (points, mean (SD))	6.9 ± 2.1	3.1 ± 2.5	**<0.001**
Endoscopic Outcomes			
Endoscopic remission (n, %)	2 (4.4)	19 (35.8)	**0.039**
Endoscopic response (n, %)		31 (63.3)	
Mucosal healing (n, %)	1 (2.3)	9 (35.8)	0.219
Mayo endoscopic subscore (points, mean (SD))	2.7 ± 0.6	2.1 ± 1.1	**<0.001**
UCEIS (points, mean (SD))	5.3 ± 1.6	4.1 ± 2.7	**0.009**
Histologic Outcomes			
Histologic remission (n, %)	0 (0)	7 (29.2)	**0.009**
Nutritional state			
ALB (g/L, mean (SD))	37.0 ± 5.8	40.3 ± 5.8	**<0.001**
ALB > 35 (n, %)	41 (63.0)	44 (86.3)	**<0.001**
Hb (g/L, mean (SD))	110.2 ± 24.0	114.7 ± 35.5	0.292
Normal Hb (n, %)	31 (43.1)	32 (56.1)	0.180
Inflammatory burden			
Normal CRP (n, %)	36 (51.4)	42 (77.8)	**<0.001**
Normal ESR (n, %)	36 (62.1)	28 (65.1)	0.180
Other Laboratory Parameters			
PLT (10^9^/L, mean (SD))	345.1 ± 96.2	334.2 ± 114.6	0.150
Normal PLT (n, %)	23 (41.1)	15 (41.7)	0.388
Normal ALT (n, %)	37 (77.1)	22 (91.7)	0.375
Normal AST (n, %)	38 (97.4)	22 (95.7)	1.000
Normal TBIL (n, %)	23 (76.7)	14 (77.8)	1.000
Normal TC (n, %)	23 (76.7)	13 (76.5)	1.000
Normal Cr (n, %)	34 (94.4)	21 (95.5)	1.000
Normal D-dimer (n, %)	6 (20.7)	8 (61.5)	1.000
Adverse Events	-	6 (8.5)	-

**UCEIS**, ulcerative colitis endoscopic index of severity; **ALB**, albumin; **Hb**, hemoglobin; **CRP**, C-Reactive Protein; **ESR**, Erythrocyte Sedimentation Rate; **PLT**, Platelets; **ALT,** Alanine Aminotransferase; **AST,** Aspartate Aminotransferase; **TBIL**, Total Bilirubin; **TC**, Total Cholesterol; **Cr**, Creatinine.

**Table 3 biomedicines-13-00190-t003:** Pre-treatment and 12-week follow-up clinical, endoscopic, and laboratory outcomes of upadacitinib treatment in patients with CD.

Outcomes	Pre-Treatment (N = 156)	12-Week Follow Up (N = 130)	*p*-Value
Clinical Outcomes			
Steroid-free clinical remission (n, %)	20 (19.4)	61 (76.8)	**<0.001**
Clinical remission (n, %)	21 (19.8)	63 (77.8)	**<0.001**
Clinical response (n, %)		64 (81.0)	
CDAI (points, mean (SD))	214.9 ± 77.9	117.5 ± 79.3	**<0.001**
Endoscopic Outcomes			
Endoscopic remission (n, %)	12 (12.6)	12 (19.4)	0.774
Endoscopic response (n, %)		23 (48.9)	
Mucosal healing (n, %)	5 (5.3)	3 (4.9)	0.250
SES-CD (points, median (IQR))	10 (5.0–18.0)	4 (1.0–8.5)	**<0.001**
Imaging Outcomes			
Radiological remission (n, %)	1 (1.1)	2 (9.1)	**<0.001**
Radiological response (n, %)		18 (85.7)	
Intestinal Doppler Ultrasound remission (n, %)	5 (3.8)	4 (5.7)	**<0.001**
Intestinal Doppler Ultrasound response (n, %)		38 (56.7)	
BWT ≤ 3 (n, %)	5 (3.6)	4 (5.6)	0.125
IUS-SAS (points, median (IQR))	72.0 (54.1–80.0)	64.7 (35.8–80.0)	**<0.001**
Perianal response			
VanAssche (points, median (IQR))	9 (0.5–16.0)	8 (4.0–13.5)	**0.023**
Nutritional state			
ALB (g/L, mean (SD))	37.3 ± 5.5	41.4 ± 5.0	**<0.001**
ALB > 35 (n, %)	94 (63.9)	91 (91.0)	**<0.001**
Hb (g/L, mean (SD))	123.9 ± 65.6	118.8 ± 22.0	0.344
Normal Hb (n, %)	85 (56.7)	61 (60.4)	0.839
Inflammatory burden			
Normal CRP (n, %)	90 (60.0)	81 (82.7)	**0.001**
Normal ESR (n, %)	75 (62.0)	55 (78.6)	0.189
Other Laboratory Parameters			
PLT (10^9^/L, mean (SD))	326.9 ± 110.4	319.3 ± 103.3	**0.023**
Normal PLT (n, %)	65 (43.6)	49 (47.1)	0.405
Normal ALT (n, %)	144 (97.3)	92 (92.9)	0.125
Normal AST (n, %)	146 (98.6)	90 (90.9)	**0.021**
Normal ALT (n, %)	114 (85.7)	70 (81.4)	0.286
Normal TC (n, %)	109 (85.2)	65 (84.4)	0.824
Normal Cr (n, %)	135 (94.4)	87 (98.9)	**0.031**
Normal D-dimer (n, %)	81 (77.9)	45 (77.6)	**0.774**
Adverse Events	-	18 (11.5)	-

**CDAI**, Crohn’s Disease Activity Index; **SES-CD**, Simple Endoscopic Score for Crohn’s Disease; **ALB,** albumin; **Hb**, hemoglobin; **CRP**, C-Reactive Protein; **ESR**, Erythrocyte Sedimentation Rate; **PLT**, Platelets; **ALT**, Alanine Aminotransferase; **AST,** Aspartate Aminotransferase; **TBIL**, Total Bilirubin; **TC**, Total Cholesterol; **Cr**, Creatinine.

## Data Availability

The datasets generated and/or analyzed during the current study are available from the corresponding author on reasonable request.

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
