# Peer review of "An Analysis of the Effectiveness and Safety of Upadacitinib in the Treatment of Inflammatory Bowel Disease: A Multicenter Real-World Study"

_biomedicines, 2025, doi:10.3390/biomedicines13010190_

Round 1

Reviewer 1 Report

Comments and Suggestions for Authors

Dear Authors of the manuscript biomedicines-3411730, thank you for this work. Before the Erudite Editor decides, I want to clarify some points with you.

1. Your abstract must be strengthened in its conclusions. Write the reasons behind your conclusions. Also, exemplify the future research Endeavors we must seek.

2. Compare the strengths of your study with the limitations of similar previous published studies in the introduction. Use the final paragraph for this. Why your study is better and must be published?

3. In the statistical analysis section, please explain your choices based on the statistical method characteristics. This will enhance the readers’ experience.

4. Please add subsections to the introduction adding details about the physiological aspects of CD and UC that involve JAK signaling. Do not forget artwork. Scientific images are welcome and highly recommended.

5. Add upadacitinib molecular and chemical details involving molecular structure, physicochemical properties, pharmacodynamics, and pharmacokinetics. Artwork for the structure is highly recommended. Also, add information about the structure’s modifications with and without the target.

6. You did not present a dedicated conclusion section. Please add a conclusions section and highlight future research endeavors in this section.

MINOR

7. There is no need for a running title.

8. The in-text references are not following MDPI’s style.

Thank you for your time and consideration.

Author Response

Comments 1: Your abstract must be strengthened in its conclusions. Write the reasons behind your conclusions. Also, exemplify the future research Endeavors we must seek.

Response 1: Thank you for pointing this out. We agree with this comment. Therefore, We have strengthened the conclusions by providing the underlying reasons for our findings and outlined specific future research endeavors that are necessary. These changes can be found in the Abstract section, specifically in the Conclusions paragraph (page 1-2, lines 42-48).

Comments 2: Compare the strengths of your study with the limitations of similar previous published studies in the introduction. Use the final paragraph for this. Why your study is better and must be published?

Response 2: Thank you for pointing this out. We agree with this comment. Therefore, We have compared the strengths of our study with the limitations of similar previously published studies in the final paragraph of the Introduction section (page 4, paragraph 2, 119-130).

Comments 3: In the statistical analysis section, please explain your choices based on the statistical method characteristics. This will enhance the readers’ experience.

Response 3: Thank you for pointing this out. We agree with this comment. Therefore, We have elaborated on the rationale behind the selection of statistical methods by explaining the characteristics of each statistical test and why they were chosen based on the data distribution and study design. These changes can be found in the 2.6 Statistical Analysis section (page 6, lines 216-235).

Comments 4: Please add subsections to the introduction adding details about the physiological aspects of CD and UC that involve JAK signaling. Do not forget artwork. Scientific images are welcome and highly recommended.

Response 4: We agree with your suggestion to add subsections detailing the physiological aspects of Crohn's disease (CD) and ulcerative colitis (UC) involving JAK signaling. In response, we have expanded the introduction to include a comprehensive overview of the JAK/STAT pathway's role in the pathophysiology of IBD. Additionally, we have incorporated a scientific image to illustrate the JAK/STAT signaling mechanism. These changes can be found on (page 2, paragraph 3, lines 68-79), and the accompanying figure is included as Figure 1(page 3) in the manuscript.

Comments 5: Add upadacitinib molecular and chemical details involving molecular structure, physicochemical properties, pharmacodynamics, and pharmacokinetics. Artwork for the structure is highly recommended. Also, add information about the structure’s modifications with and without the target.

Response 5: We agree with your suggestion to include detailed molecular and chemical information about upadacitinib, as well as visual artwork depicting its structure.  Therefore, we have expanded the introduction to incorporate comprehensive details regarding upadacitinib’s molecular structure, physicochemical properties, pharmacodynamics, and pharmacokinetics. Additionally, we have included a figure illustrating the molecular structure of upadacitinib and its structural modifications with and without the target.

Regarding your request for information about the structure’s modifications with and without the target, we would like to note that upadacitinib does not undergo significant structural modifications upon binding with its target (JAK1). Therefore, we have not included any structural modifications in our revision.

These changes can be found on (page 3, paragraph 1, lines 86-93) and the accompanying artwork is provided as Figure 2 in the manuscript.

Comments 6: You did not present a dedicated conclusion section. Please add a conclusions section and highlight future research endeavors in this section.

Response 6: 

Thank you for pointing this out. We agree with this comment. Therefore, We have added a dedicated Conclusions section that summarizes the study findings and highlights future research endeavors. This change can be found at the end of the manuscript as section 5, titled "Conclusions."(page 19, lines 483-496)

Comments 7(MINOR): [There is no need for a running title.]

Response 7: 

Thank you for pointing this out. We have removed the running title from the manuscript.

Comments 8(MINOR): [The in-text references are not following MDPI’s style.]

Response 8: 

Thank you for pointing this out. We have revised all in-text references to follow MDPI’s numerical citation style using square brackets.

Reviewer 2 Report

Comments and Suggestions for Authors

The authors claimed that upadacitinib effectively treated both UC and CD in a Chinese cohort, demonstrating significant improvements across multiple disease parameters, which helps endorse upadacitinib's effectiveness and rapid onset in treating refractory IBD.

Major:

1.      In Fig. 1, there is no description or explanation provided for the exclusion of the 18 patients.

2.      In Fig. 1, there is no data for female patients. Why is this the case? There is no description or explanation provided for this omission.

  1. In “2.3. Investigated Drugs”: the protocol recommended induction dose from this link RINVOQ® (upadacitinib) Dosing for Ulcerative Colitis and Crohn's Disease (see link below) is as follows:
    1. CD: 45 mg once daily for 12 weeks
    2. UC: 45 mg once daily for 8 weeks.

Please ensure that "once daily" is not omitted.

The correct link is https://www.rinvoqhcp.com/gastroenterology/dosing-lab-monitoring . However, it is better to cite a reference paper for reference 7 and 8, as the link may become invalid in the future.

Minor:

1.      The caption of Table 3 should be "12-week follow-up" instead of "8-week follow-up."

2.      Maybe it is better that In Fig. 1, include the numbers of "CD Patients (N = 156)" and "UC Patients (N = 80)" to specify the number of patients in each category.

3.      Line 335, Is reference 14 the best one? Here is the good one: PMID: 35644166.

4.      Line 336, week 8 (induction) and week 52 (maintenance).

5.      Line 405, is this study the first study from China about real-world use? Here are two other papers:

https://doi.org/10.1007/s10238-024-01468-z

https://doi.org/10.1093/ecco-jcc/jjad212.0768

Author Response

Comments 1-2:

1. In Fig. 1, there is no description or explanation provided for the exclusion of the 18 patients.;

2. In Fig. 1, there is no data for female patients. Why is this the case? There is no description or explanation provided for this omission.

Response 1-2: Thank you for pointing this out. We agree with this comment. Therefore, I have updated Figure 3 with a more detailed description of the exclusion reasons for the 18 patients. Additionally, We have clarified that gender data is already described in the Results section and Table 1, hence it is not included in Figure 3. These changes can be found in the manuscript’s patient selection paragraph and in Figure 3s flow lines (page 7, lines 242-245).

Comments 3: In “2.3. Investigated Drugs”: the protocol recommended induction dose from this link RINVOQ® (upadacitinib) Dosing for Ulcerative Colitis and Crohn's Disease (see link below) is as follows:

CD: 45 mg once daily for 12 weeks

UC: 45 mg once daily for 8 weeks.

Please ensure that "once daily" is not omitted.

The correct link is https://www.rinvoqhcp.com/gastroenterology/dosing-lab-monitoring . However, it is better to cite a reference paper for reference 7 and 8, as the link may become invalid in the future.]

Response 3: Thank you for pointing out the omission of "once daily" in the dosing regimen and for recommending the replacement of the external link with reference papers to ensure the longevity of our citations. We agree with your comments and have made the necessary revisions to enhance the clarity and reliability of our manuscript. Specifically, we have cited the FDA-issued dosing guidelines and the correct website link provided to ensure the dosing regimen is both accurate and credible. These changes can be found in section 2.3 "Investigated Drugs" (page 4, lines 156-159).

Comments 4(Minor:): The caption of Table 3 should be "12-week follow-up" instead of "8-week follow-up."

Response 4: Thank you for pointing out the discrepancy in the caption of Table 3. We agree with your comment and have made the necessary revision. Therefore, we have updated the caption to "12-week follow-up" as suggested(page 15, lines 365).

Comments 5(Minor:): Maybe it is better that In Fig. 1, include the numbers of "CD Patients (N = 156)" and "UC Patients (N = 80)" to specify the number of patients in each category.

Response 5: Thank you for your suggestion to include the numbers of "CD Patients (N = 156)" and "UC Patients (N = 80)" in Fig. 1. We agree with your comment, and as a result, we have made the necessary revisions. Additionally, due to the inclusion of new content, the figure number has been updated to Fig. 3. This change can be found in Figure 3 of the revised manuscript (page 7).

Comments 6(Minor:): Line 335, Is reference 14 the best one? Here is the good one: PMID: 35644166.]

Response 6: Thank you for your comment regarding the reference in Line 335. We agree with your suggestion, and have replaced reference 14 with the recommended one (PMID: 35644166).This change can be found in Line 410 of the revised manuscript.

We appreciate your careful review and helpful suggestion.

Comments 7(Minor:):  Line 336, week 8 (induction) and week 52 (maintenance).

Response 7: Thank you for pointing this out. We agree with your comment and have corrected the text. The revised version now correctly states "week 8 (induction) and week 52 (maintenance)."

This change can be found in Line 388 of the revised manuscript.

Comments 8(Minor:): Line 405, is this study the first study from China about real-world use? Here are two other papers:

https://doi.org/10.1007/s10238-024-01468-z

https://doi.org/10.1093/ecco-jcc/jjad212.0768

Response 8: 

Thank you for your insightful comments.

We appreciate your observation regarding the presence of other real-world studies from China. We agree that the studies you referenced are valuable contributions to the field.  However, our study distinguishes itself in several key aspects.

Reference 1 (DOI: https://doi.org/10.1007/s10238-024-01468-z) is a single-center case series focusing exclusively on 14 patients with acute severe ulcerative colitis (ASUC).  Its limited sample size and narrow focus on ASUC without including Crohn's disease patients restrict its generalizability and comprehensive assessment of upadacitinib's effectiveness across the broader spectrum of IBD.

Reference 2 (DOI: https://doi.org/10.1093/ecco-jcc/jjad212.0768) presents real-world data from China with a larger sample size;  however, it primarily concentrates on Crohn's disease patients and does not extensively investigate ulcerative colitis.  Additionally, as a conference abstract, the study's results have not undergone full peer review or detailed publication, limiting the depth of available information.

In contrast, our multicentric retrospective cohort study encompasses a larger and more diverse patient population, including both Crohn's disease and ulcerative colitis patients.  This comprehensive approach allows for a more robust analysis of upadacitinib's effectiveness and safety across different IBD phenotypes.  Furthermore, our study integrates multifaceted assessment methods, including endoscopic, imaging, histological, and laboratory parameters, providing a holistic evaluation of treatment responses that surpasses the scope of the aforementioned studies.

We appreciate your attention to detail and your valuable feedback.

Round 2

Reviewer 1 Report

Comments and Suggestions for Authors

Thank you for your time and consideration.